# Amino Acid Compound 2 (AAC2) Treatment Counteracts Insulin-Induced Synaptic Gene Expression and Seizure-Related Mortality in a Mouse Model of Alzheimer’s Disease

**DOI:** 10.3390/ijms252111689

**Published:** 2024-10-30

**Authors:** Zhijie Deng, Aejin Lee, Tao Lin, Sagarika Taneja, Devan Kowdley, Jacob H. Leung, Marykate Hill, Tianyi Tao, Julie Fitzgerald, Lianbo Yu, Joshua J. Blakeslee, Kristy Townsend, Zachary M. Weil, Jon R. Parquette, Ouliana Ziouzenkova

**Affiliations:** 1Department of Human Sciences, The Ohio State University, Columbus, OH 43210, USA; deng.909@buckeyemail.osu.edu (Z.D.); or ajlee@mju.ac.kr (A.L.); kowdley.2@buckeyemail.osu.edu (D.K.); leung.167@buckeyemail.osu.edu (J.H.L.); hill.2343@buckeyemail.osu.edu (M.H.); 2Department of Food and Nutrition, Myongji University, 116 Myongji-ro, Cheoin-gu, Yongin-si 17058, Gyeonggi-do, Republic of Korea; 3Department of Chemistry and Biochemistry, The Ohio State University, Columbus, OH 43210, USA; lin.1338@buckeyemail.osu.edu (T.L.); taneja.33@buckeyemail.osu.edu (S.T.); parquette.1@osu.edu (J.R.P.); 4Department of Neurological Surgery, The Ohio State University College of Medicine, Columbus, OH 43210, USA; tianyi.tao@osumc.edu (T.T.); kristy.townsend@osumc.edu (K.T.); 5Department of Neuroscience, The Ohio State University, Columbus, OH 43210, USA; julie.fitzgerald@osumc.edu (J.F.); zachary.weil@hsc.wvu.edu (Z.M.W.); 6Department of Biomedical Informatics, The Ohio State University, Columbus, OH 43210, USA; lianbo.yu@osumc.edu; 7Department of Horticulture and Crop Science, Ohio Agricultural Research and Development Center (OARDC), The Ohio State University, Columbus, OH 43210, USA; blakeslee.19@osu.edu; 8Department of Neuroscience, WVU Rockefeller Neuroscience Institute, West Virginia University, Biomedical Research Center (BMRC), Morgantown, WV 26506, USA

**Keywords:** Alzheimer’s disease, dementia, epilepsy, diabetes, synapses, cognitive, seizures, nanomaterials, dipeptides, nanofibers, insulin

## Abstract

Diabetes is a major risk factor for Alzheimer’s disease (AD). Amino acid compound 2 (AAC2) improves glycemic and cognitive functions in diabetic mouse models through mechanisms distinct from insulin. Our goal was to compare the effects of AAC2, insulin, and their nanofiber-forming combination on early asymptomatic AD pathogenesis in APP/PS1 mice. Insulin, but not AAC2 or the combination treatment (administered intraperitoneally every 48 h for 120 days), increased seizure-related mortality, altered the brain fat-to-lean mass ratio, and improved specific cognitive functions in APP/PS1 mice. NanoString and pathway analysis of cerebral gene expression revealed dysregulated synaptic mechanisms, with upregulation of *Bdnf* and downregulation of *Slc1a6* in insulin-treated mice, correlating with insulin-induced seizures. In contrast, AAC2 promoted the expression of *Syn2* and *Syp* synaptic genes, preserved brain composition, and improved survival. The combination of AAC2 and insulin counteracted free insulin’s effects. None of the treatments influenced canonical amyloidogenic pathways. This study highlights AAC2’s potential in regulating synaptic gene expression in AD and insulin-induced contexts related to seizure activity.

## 1. Introduction

Half a century of global obesity [1] and diabetes [2] epidemics [1,2] escalate risks for neurodegenerative diseases, including Alzheimer’s disease (AD) dementia [3,4]. An alarming 81% of patients with AD have elevated fasting glucose or type 2 diabetes mellitus (T2DM) [5,6,7]. The systemic and/or cerebral occurrence of the diabetic quadruplet conditions—systemic hyperglycemia, cerebral hypoglycemia, insulin resistance, and hyperinsulinemia—concurrently influence the risk of AD (reviewed in [8]), as well as clinical AD pathogenesis, including chronic inflammation and vascular pathology [9], reduced cognitive performance, and epileptic activity [10]. These changes precede the development of AD hallmark pathology, such as the accumulation of neurofibrillary tangles (NFT) and amyloid beta (Aβ) plaques in AD brains [9]. The canonical Aβ and NFT pathology in AD patients with T2D, compared to those without this disease, was reported to be similar in some studies within general AD populations [6] but differs in genetically predisposed AD carriers [9]. Thus, the diabetic quadruplet conditions appear to facilitate early processes leading to the development of AD [11,12,13,14].

Insufficient glucose supply to the brain, i.e., hypoglycemia, has been proposed as a central initiating factor in Alzheimer’s disease (AD) pathology, triggering a shift from physiological to maladaptive responses that ultimately contribute to the progression of AD [15], while hyperglycemia in the peripheral circulation has long-term consequences for cerebral oxidative stress, inflammation, and hypoxic metabolism. The diminished cerebral intracellular glucose uptake disrupts connectivity between different brain regions in subjects with mild cognitive impairment, which is exacerbated in AD patients [16]. This transition is pivotal in defining the disease’s trajectory [17]. Cerebral hypoglycemia can rapidly alter synaptic excitation and inhibition (E/I) balance [17], neuronal polarization, and synaptic activity [18], which may generate intrinsic epileptiform activity or seizures [19] and is correlated with the progression and mortality of AD [10,20].

The management of systemic and cerebral hyperglycemia and insulin resistance includes insulin or insulin-centric pharmaceuticals [21], although transient hypoglycemic episodes in the brain remain a major side effect of insulin therapies [21], stemming from distinct differences in the regulation of glucose uptake by insulin via the glucose transporter GLUT4 in peripheral versus cerebral tissues, which rely on GLUT1 and GLUT3. Cerebral hypoglycemia induced by insulin, in severe cases, can lead to seizures, coma, and death [21,22,23]. Despite these challenges, insulin treatment in AD patients is considered as a strategy to improve cognitive performance [22]. To refine glycemic control and avoid the risks of insulin therapy alone, we developed an amino acid compound 2 (AAC2) comprised of a lysine dipeptide with a coumarin side chain [24]. AAC2 binds atypically with the leptin receptor, facilitating GLUT1-dependent glucose uptake, which leads to significantly improved systemic and potentially cerebral glycemic control [24,25]. This interaction also results in cerebral effects, including enhanced cerebral glucose uptake ex vivo [26] and reduced anxiety in mice with experimental T1DM in vivo [24,25]. AAC2 self-assembles into positively charged nanofibers and electrostatically binds negatively charged insulin. This AAC2–INS complex prevents hypoglycemic episodes and cognitive dysfunction observed in untreated diabetic mice or those treated with free insulin [25]; however, its efficacy in decreasing AD progression and potential side effects, given that both natural and synthetic fibrils carry the risk of exacerbating amyloidosis (reviewed in [27]), have not been examined.

The aim of our study was to compare the pathophysiologic, genetic, and cognitive effects of classical insulin to AAC2 and AAC2–INS nanofibers in a mouse model of AD pathogenesis. Given that addressing glucose metabolism dysfunction in the early stages of AD may delay or prevent disease progression [28,29], we included insulin as the positive control group. Based on a report that male APP/PS1 mice demonstrate impaired glucose metabolism in the glucose tolerance test as early as 2 months of age, prior to amyloid beta deposition and cognitive decline at 8–9 months of age [30], we treated male APP/PS1 mice before the onset of this AD pathogenesis (Figure 1a). In this probe trial study, we included between six and nine mice per group, which is comparable to the number used in a previous study that performed behavioral tests with five to ten APP/PS1 mice per group [31].

## 2. Results

The effects of insulin (I), AAC2 (A), and nanofiber complex (A+I) were studied in young APP1/PS1 male mice (Figure 1a) to assess early and signature pathophysiological and behavioral changes in AD pathogenesis. The control WT male group was used to establish the physiological range of measured parameters compared to treated and untreated APP1/PS1 mice. The treatment regimen was based on the efficacy of these compounds in mouse models of diabetes [24,25], while a study duration of 17 weeks allowed observation of early pathology without distinct cognitive abnormalities of AD between APP1/PS1 and WT mice [32,33].

### 2.1. Normal Peripheral Glycemic Control but Altered Mortality in APP/PS1 Mice

The APP/PS1 mice, which had similar weight (Figure 1b) and fasting glucose(Figure 1c, Day 0), were randomized among control and treated group. The fasting glucose levels (Figure 1c) in all groups were within a normal non-diabetic range throughout this study. Although all mice maintained similar weight and peripheral glycemic control throughout this study, the survival varied among the treated APP/PS1 mice (Figure 1d). One mouse per group died of undetermined reasons in the control and two in the nanofiber (A+I) APP/PS1 group. Four out of seven treated APP/PS1 mice from the I group developed seizures and died immediately after injection of insulin in the course of study. No mortality was observed in APP/PS1 mice treated with AAC2 and in the WT mice. Thus, survival in APP/PS1 mice was decreased by insulin to 57%, whereas AAC2-treated mice showed a survival rate of 100%. Notably, insulin bound with AAC2 reduced incidence of death in A+I nanofiber group to the levels seen in control APP/PS1 mice, suggesting that nanofibers or a combination of INS with AAC2 can alter responses to insulin.

### 2.2. Similar Metabolic Parameters and Activity within Treated APP/PS1 Mice

Metabolic parameters were assessed in single-housed APP/PS1 mice during 24 h at the end of the study between the 16th and 17th weeks of treatment (Figure 2). The overall energy expenditure was similar in APP/PS1 mice with and without treatment (Figure 2a–c). The overall respiratory exchange rate (RER) kinetics are shown in Figure 2d. RER was significantly lower in all treated compared to untreated groups of APP/PS1 mice during the resting light cycle, demonstrating the increased utilization of glucose as energy substrate in untreated APP/PS1 mice (Figure 2e). During the active dark cycle, RER was similar among all groups (Figure 2f). The XYZ activity kinetics exhibited a specific pattern compared to energy expenditure and RER (Figure 2g) due to a low activity during the light period (Figure 2h) and its marked increase during the dark cycle in all mouse groups (Figure 2i). Nonetheless, these activity patterns were not statistically different among all groups of APP/PS1 mice (Figure 2j). Thus, AAC2, insulin, and their nanofiber complex exerted similar influence on the basal metabolic responses.

### 2.3. Similar Body and Different Brain Composition in Treated APP/PS1 Mice

We employed EchoMRI for an objective analysis of the whole body and frozen brain tissues in APP/PS1 mice (Figure 3). The reference physiological values for the healthy mice from this genetic background were obtained from the WT group. All studied mice had similar weight, lean, and fat mass at the end of the study (Figure 3a–c). Notably, while lean mass was nearly identical in WT and APP/PS1 mice (Figure 3b), the mass of fat in APP/PS1 mice was 192.6% higher compared to fat mass seen in the WT group (100%) (Figure 3c). All treated and untreated APP/PS1 mice had comparable lean and fat mass (Figure 3b,c). To account for variability in the weight of the animals, we also compared the lean and fat composition as a percentage of mouse body weight (Figure 3d–f). Our data indicate a minor reduction in lean (Figure 3e,f) and an increase in fat proportion (Figure 3f) in APP/PS1 mice from all groups compared to the WT mice. These data demonstrate minor differences in body composition between WT and APP/PS1 genotype, which were not influenced by the type of treatment.

In contrast, the brain composition was affected by treatment (Figure 3g). The brain mass in AAC2-treated APP/PS1 mice was higher than in the untreated APP/PS1 group due to increased lean (Figure 3h)—but not fat—mass (Figure 3i). In all APP/PS1 and WT mice, the lean and fat mass was similar. A comparison of the proportion of lean and fat mass normalized to brain weight indicated a significant shift in brain composition in INS-treated APP/PS1 mice compared to the control group (Figure 3j). The change in the INS-treated group was due to the moderate increase in lean (+5.2%, Figure 3k) and the marked decrease (−30%, Figure 3l) in fat proportions compared to untreated APP/PS1 mice (100%). The changes in the other treatment groups were not significant. These results indicated the predominant effect of insulin and AAC2 on the brain, leading to changes in the brain composition, whereas peripheral tissue underwent minor changes in the early phase of AD pathogenesis in APP/PS1 mice.

### 2.4. Similar Cognitive Performance in WT and APP/PS1 Mice but Improved Performance in INS Compared to AAC2–INS-Treated APP/PS1 Mice

Although APP/PS1 were in the early phase of AD pathogenesis, we examined the behavioral patterns and cognitive performance in APP/PS1 vs. WT mice (Figure 4). An open field test was performed 11–12 weeks after the beginning of treatment (Figure 4a–c). The total and peripheral distances were similar in WT and all groups of APP/PS1 mice (Figure 4a,b).

The animals predominantly had peripheral movement. There was no statistical difference in the number of rears among all mouse groups (Figure 4c). In each group, mice exhibited large variability in their behavior. All WT and APP/PS1 mice spent analogous amounts of time on the rod in the rotarod test (Figure 4d). Anxiety-related behavior was also examined using an elevated plus-maze test 14–15 weeks after the beginning of treatment (Figure 4e,f). All WT and APP/PS1 mice spent similar amounts of time in the open (Figure 4e) and closed arm (Figure 4f) in the maze. Both tests indicated comparable anxiety-related behavior in WT and APP/PS1 mice with and without 15 weeks of treatment.

The cognitive performance assessed by Barnes maze 14–15 weeks after the beginning of treatment (Figure 4g–k) revealed no significant difference in all measures of the special learning and memory abilities between WT and any group of APP/PS1 mice with and without treatment in 5–6-month-old mice. The reported changes of cognitive decline in APP/PS1 mice occurs after 6 months of age [36] and were distinct at 9 months of age [37]. Given the continuous mortality in insulin-treated mice, our study lasted 17 weeks and ended at around 5–6 months of age. However, we found that specific learning characteristics were significantly different among treatment groups. During the training phase, all APP/PS1 mice groups covered similar distances to the escape hole (Figure 4g), but mice treated with free insulin made significantly less errors compared to APP/PS1 mice treated with AAC2 or nanofiber AAC2–INS (Figure 4h). These findings corroborate the results in the testing phase (Figure 4i–k). The distance measure did not change among all APP/PS1 groups (Figure 4i). Insulin-treated APP/PS1 mice also spent a higher percentage of time in the area missing the escape box compared to AAC2–INS-treated mice (Figure 4j). The number of errors was lower in mice treated with free insulin compared to those treated with AAC2–INS nanofibers (Figure 4k). Thus, paradoxically, the surviving APP/PS1 mice treated with insulin appear to retain spatial memory to a larger extent than mice treated with AAC2–INS nanofibers, even though these mice exhibit an improved survival profile. Therefore, we performed cerebral gene expression analyses to identify early changes in AD pathogenesis and mechanisms underlying the differences in survival rate and behavioral test results among free insulin, AAC2, and their nanofiber AAC2–INS combination.

### 2.5. Increased Expression of Canonical AD Genes in APP/PS1 Mice with and Without Treatment Compared to WT Mice

We employed quantitative NanoString neurometabolic analysis [38] to examine expression of 99 cerebral genes, including canonical AD genes (*App* and *Psen2*) [39], neurometabolic genes, as well as genes regulated by AAC2 in diabetes [25], in WT and all APP/PS1 mice (Figure 5 and Figure 6). To elucidate the relative impact of treatment on canonic AD vs. other pathways, we created a scatter plot of the log fold change [40]. As expected, the log comparison of cerebral genes was different in WT and control APP/PS1 mice, reveling a significantly higher expression of canonic *App* and relative to all treatments (log comparison to APP/PS1 control, Figure 5a–c). The KEGG pathway analysis of DEGs between WT and control APP/PS1 mice revealed AD and neurodegeneration as the principal pathways altered in APP/PS1 mice (Figure 5d). These findings validated the onset of AD pathogenesis in all APP/PS1 mice vs. WT.

In addition, the scatter plots revealed fundamentally similar patterns in the expression of genes downstream of prion formation, such as phosphoinositide-dependent kinase-1 (*Pdk1*) [41], and genes defining the permeability of the blood–brain barrier (BBB) plasmalemma vesicle-associated protein (*Plvap*) [42] and claudin 5 (*Cldn5*) [43], comparing WT to all groups of APP/PS1 mice (Figure 5a–c). Nevertheless, we found minor but significant differences in the regulation of *App* within APP/PS1 treatment groups (Figure 5e), which was lower in INS-treated compared to AAC2- or AAC2–hINS nanofiber-treated mice. Also, none of these genes was different compared to control APP/PS1 mice. The other *Psen2*, *Pdk1*, and *Plvap* gene expression regulated downstream of *App* was not significantly influenced by treatment (Figure 5f–h). Consistent with a phenotype [44], *Psen2* expression was significantly decreased in APP/PS1 control mice compared to WT.

The scatter plot also revealed that AAC2, INS, or AAC2–INS nanofibers regulate a specific set of genes, including carbonic anhydrase 14 (*Car14*) and beta-glucuronidase (*Gusb*), responsible for drug metabolism (Figure 5a–c). Cumulatively, the expression of genes supports the onset of AD pathogenesis in APP/PS1 mice in our study. However, none of the treatments affected the expression of these canonical AD genes.

### 2.6. Distinct Regulation of Neuroinflammatory Genes by INS vs. AAC2 or AAC2–INS Nanofibers in APP/PS1 Mice

To discover the principal pathways targeted by INS and AAC2 treatments, we performed Reactome pathway over-representation analysis based on cerebral gene expression analysis in mice from all groups. The Reactome pathway analysis of DEGs between AAC2 and hINS treatment is shown in Figure 6a. This analysis validated changes of known INS pathways termed ‘The regulation of INS-like growth factor transport and uptake by insulin-like growth factor binding proteins’. The AAC2 treatment led to the anticipated changes in extracellular matrix pathways, peptide ligand binding receptors, and extracellular matrix organization consistent with their potential properties to form nanofiber or bind electronegative proteins or peptides.

The GPCR pathways in our study included genes such as *S100B*, *Nos1*, *Ccl5*/RANTES, and *Ccl3/MIP1a* (Figure 6b–e), which have previously been associated with neuroinflammation in AD [45,46,47] and epileptic activity [48]. AAC2 treatment increased *Nos1* expression (Figure 6b) but decreased expression of *S100b* (Figure 6c) compared to all other animal groups. With the exception of *Ccl3* (Figure 6d), which was higher in APP/PS1 control vs. WT mice, these neuroinflammatory genes were similar in WT and untreated APP/PS1 control mice. On the other hand, INS- and AAC2-treated APP/PS1 mice exhibited different inflammatory patterns. INS treatment resulted in the increased expression of cytokine *Ccl5*/RANTES (Figure 6e) but decreased *Ccl3/MIP1a* expression compared to AAC2-treated mice, AAC2–INS nanofibers, and WT groups. Nevertheless, INS and AAC2 both influence inhibitory and activating components of neuroinflammation, likely balancing these responses. Neither INS nor AAC2 reduced *Ccl3* expression seen in untreated APP/PS1 control mice. This modest regulation of inflammation and its balancing patterns may not be the mechanism driving divergent responses to INS and AAC2 in APP/PS1 mice.

### 2.7. Opposite Regulation of Genes Determining Synaptic Excitation/Inhibition Balance by INS vs. AAC2 or AAC2–INS Nanofibers

Notably, the major other pathways defining the difference between responses to INS and AAC2 were related to synaptic function (synaptophysin (*Syp*, NM_009305.2), synapsin II (*Syn2*, NM_013681.3)*,* brain-derived neurotrophic factor (*Bdnf*), and sodium-dependent neurotransmitter transporter 1 (*Slc6a1*), Figure 7a–d). These genes mediating GABAergic signaling were previously implicated in the pathophysiology of AD and epilepsy due to their role in synaptic function [49,50,51,52,53,54].

The synaptic inhibition genes *Syp* (Figure 7a) and *Syn2* (Figure 7b) had the highest levels of expression in AAC2-treated mice among all APP/PS1 groups. *Syn2* expression levels in AAC2-treated APP/PS1 mice were also markedly higher than that in WT control. INS treatment uniquely increased expression of *Bdnf*, regulating excitatory and inhibitory GABA signaling, synaptogenesis, and neuroplasticity [55], compared to all APP/PS1 groups or WT mice (Figure 7c). Excitatory or inhibitory effects of BDNF/GABA depend on GABA transporter GAT-1 encoded by *Slc6a1*, in which suppression of *Slc6a1* supports excitatory GABA responses related to epilepsy in AD [56]. The expression of *Slc6a1* was significantly decreased in INS-treated APP/PS1 mice compared to AAC2- and AAC2–INS-treated APP/PS1 groups and WT mice (Figure 7d). The gene expression analysis demonstrates the opposite regulation of synaptic gene by INS and AAC2, consistent with a previous report on their opposite effects on GLUT1 glucose uptake in human brain barrier endothelial cells [24] in agreement with earlier findings on the dependence of synaptic inhibition in cortical pyramidal neurons and thalamic relay neurons on this pathway [57].

## 3. Discussion

In this exploratory study using a non-diabetic model of AD, we demonstrated distinct differences in cerebral response between canonical INS and AAC2, which have been previously reported in animal models of T1DM and T2DM [24,25]. Our initial objective was to compare the effects of INS versus AAC2—with and without INS—on physiological and cognitive traits, as well as cerebral gene expression, during the early, asymptomatic stage of AD pathogenesis. However, our research yielded some unexpected findings.

The first unexpected result of our study was the distinct incidence of death in response to treatment with INS vs. AAC2. Early death has been described in APP/PS1 mice, affecting approximately 40% of their population between 20 and 400 days of life [58]. This incidence in the untreated APP/PS1 mice was corroborated in our study. However, injections of INS resulted in a 40% increased incidence of mortality compared to untreated controls and were accompanied by seizures post-INS injection, leading to immediate death. This seizure-related mortality was prevented when INS was bound with AAC2, suggesting that the formation of nanofibers or the combined use of INS and AAC2 alters the response to the same dose of INS. Strikingly, no incidence of death occurred in AAC2-treated APP/PS1 mice.

Although increased neuronal excitability and elevated epileptiform activity have been observed in AD patients [59] and APP/PS1 mice [60], along with hyperexcitability [61] and seizures reported as side effects of INS [62], the seizures observed in this study were unexpected. Since the insulin treatment regimen did not induce hypoglycemic episodes in our previous studies [24,25], seizures and epileptiform activity were not anticipated and, thus, were not recorded. This represents a limitation of our study, warranting further investigation with a larger cohort of animals. Nevertheless, our results suggest that INS may increase the risk of AD-related pathologies, such as seizures and possibly epileptiform activity, which may have contributed to the unexpected deaths in the INS-treated mice. This side effect of INS treatment could potentially be mitigated by the formation of AAC2–INS nanofibers or the use of AAC2 alone, as demonstrated in this exploratory study. Intranasal INS therapy has been increasingly applied for the treatment of mild cognitive impairments in AD patients [22,63], with controversial outcomes reported in larger studies [64]. In our study, we also found that surviving mice treated with INS made significantly fewer errors in the Barnes maze test compared to other treatment groups, suggesting a delay in the onset of some cognitive deficits in INS-treated mice. This beneficial effect was abolished when INS was bound to AAC2, strengthening the observation that the properties of AAC2–INS nanofibers differ from those of their constituents.

Comprehensive studies of the cognitive effects of INS, AAC2, and their nanofibers were limited by the short (6-month) duration of this study due to the high incidence of mortality in the INS-treated group. This time period was not sufficient to develop significant cognitive abnormalities in APP/PS1 mice compared to age-matched control WT mice, even though all APP/PS1 mice expressed significantly higher levels of *App*. These observations confirm data from previous studies showing that APP/PS1 mice develop cognitive deficits after 6 months of age [32,33]. Although functional behavioral studies are established as the gold standard for assessing drug efficacy, most of the cognitive deficits occur in the late stage of AD, triggered by Aβ deposits and tau NFT aggregates implicated in hyperexcitation [56] and/or impaired synaptic inhibition [65,66], neuroinflammation [8], and an increase in pro-apoptotic metalloproteinases [67], among other mechanisms. We studied the mechanisms of the early period of AD, characterized by nonconvulsive epileptiform activity [68] associated with impaired cerebral insulin signaling and an imbalance in cerebral glucose concentrations [23,69], but asymptomatic for cognitive deficits.

Pathway insights into the distinct cerebral responses were gained through a systematic quantitative analysis of the expression of 90 genes related to canonical Alzheimer’s disease, neurometabolic, and neuroinflammatory pathways in the brains of WT and APP/PS1 mice. The quantitative analysis of gene expression by NanoString is a reliable tool for identifying candidate pathways [38,70]: however, it provides limited insights into the signal transduction underlying epileptiform and/or synaptic activity in response to INS, AAC2, and their combination. Nonetheless, such critical features of AD as increased *APP* and decreased *Psen2* expression in APP/PS1 mice validated the onset of AD pathology in our study. A similar expression profile has recently been reported in association with chronic evoked seizures in young pre-symptomatic APP/PS1 mice [44]. The pathway analysis revealed that AD and neurodegenerative pathways were the most distinct between APP/PS1 and WT mice in our study. However, none of the treatments influenced the canonical AD genes compared to untreated APP/PS1 mice. Although the complex etiology of human AD has not been fully replicated in any mouse model [71], our findings exclude amyloidogenic processes and canonical AD pathways as primary targets of AAC2 and INS therapy.

The second unexpected outcome in our study was the absence of differences in the expression of neurometabolic genes PDK1/Akt/TACE signaling axis [72] among treatment groups, e.g., *Pdk1* expression was markedly higher in all APP/PS1 mice compared to WT mice. Thus, despite the differences in INS-dependent AKT signaling and AAC2-dependent PKC zeta activation [24,25], these pathways did not contribute additionally to *Pdk1* expression, which remains similar in all APP/PS1 mice.

Neuroinflammation appeared to be modest at the early stages of AD pathogenesis, with only the astrocyte marker *Ccl3* (alias *MIP1-α*) gene expressed at higher levels in APP/PS1 mice compared to WT mice. CCL3 also has a metabolic and cerebral role linked to insulin metabolism in T1D [73] and is associated with reduced basal synaptic transmission and impaired spatial memory when hippocampal CCL3 levels are elevated in WT mice [74]. In our study, the lower CCL3 expression in INS-treated APP/PS1 mice compared to the other treatment groups could potentially contribute to the improved cognitive performance in this group compared to the AAC2 and AAC2–INS treatments. In contrast, in our study, INS treatment significantly increased *Ccl5* (also known as RANTES) expression, which was associated with an increase in seizure-related deaths in the INS-treated group. This finding is consistent with previous reports identifying CCL5 as a marker for the late phase of experimental epilepsy [75] and epilepsy induced by autoantibodies [76]. Thus, *Ccl3* and *Ccl5* expression likely highlights the divergent neurodegenerative epileptogenic responses to AAC2 and INS in this study.

The expression levels of the canonical AD gene, *App*, were used as a marker for potential amyloidogenic effects of AAC2 or AAC2–INS nanofibers. However, *App* expression was not different in all treatment groups compared to the untreated control, even though moderately lower *App* expression was observed in the INS group compared to the AAC2 and AAC2–INS treatment groups, where the binding of INS with AAC2 counteracted the effects of INS. These data, combined with the neuroinflammatory marker profile, suggest that neither AAC2 nor AAC2–INS nanofibers promote amyloidogenic and proinflammatory genes in AD pathogenesis beyond those seen in untreated APP/PS1 mice.

The pathway analysis of cerebral gene expression revealed that synaptic E/I balance is the pivotal pathway responsible for cerebral differences in INS and AAC2 treatments. INS treatment increased *Bdnf* expression and decreased *Slc6a1* expression, a pattern that has been well-documented in the context of synaptic excitation [77,78,79,80] (Figure 7e). The change in the expression of both *Bdnf* and *Slc6a1* provide insights into the longstanding controversy regarding the role of BDNF in seizures [81] through the TrkB/GABA_A_ receptor [77,82] and cognitive performance [80] via binding to the p75 neurotrophin receptor [77]. BDNF-dependent mechanisms should be further elucidated in the future, as they likely contribute to the improved performance observed in INS-treated APP/PS1 mice.

The suppression of the *Slc6a1* gene, which encodes the GAT-1 transporter, in INS-treated APP/PS1 mice is associated with impaired GABA and taurine reuptake by astrocytes, contributing to the BDNF/TrkB/GABAergic excitatory extrasynaptic response [78] and disrupted E/I balance in AD [56]. *Slc6a1* is widely recognized as a gene associated with epilepsy and neurodegeneration in adults [83], as well as in children with the G443D genetic variant of *Slc6a1* [84]. In agreement with the excitatory gene expression profile, our study found a significant decrease in brain lipid content only in APP/PS1 mice treated with INS. Together, the gene expression patterns of *Bdnf* and *Slc6a1* suggest an excitatory imbalance, which aligns with functional outcomes such as the high incidence of seizure-related deaths in INS-treated APP/PS1 mice.

In contrast, AAC2 treatment promoted the expression of inhibitory synaptic genes, *Syn2* (synapsin II) and *Syp* (synaptophysin), examined in the context of vesicular trafficking of neurotransmitters [85], and mutation (T198I) is associated with epilepsy and intellectual disability [86]. Genetic *Syn2* deficiency in mice resulted in generalized epileptic seizures [54]. *Syn* expression has been examined in a kainic acid-induced model of epilepsy, demonstrating marked suppression after the induction of epilepsy and an increase in the late phase, suggesting a role for *Syp* in the resolution of cerebral damage [87]. Notably, these inhibitory pathways were not different between WT and APP/PS1 mice, but the levels of *Syp* and *Syn2* were significantly increased in the AAC2-treated group compared to untreated APP/PS1 mice. It is well established that the inhibition of GABAergic neurons depends on the leptin receptor (LepR) [88]. Moreover, chronic leptin administration activates ERK1/2 in neonatal rats [89]. Therefore, activation of the LepR/ERK1/2 pathway by AAC2 [24] could trigger an inhibitory response in these neurons through increased expression of *Syp* and *Syn2*. Although mechanistic studies were beyond the scope of this paper, the role of AAC2/LepR signaling in the induction of *Syp* and *Syn2* appears to be a plausible mechanism supporting inhibitory synaptic balance and preventing early death in APP/PS1 mice. Regardless of interpretation, our results revealed that AD pathology could increase the risk of side effects of INS, which could be reduced by the formation of AAC2–INS nanofibers. The potential of AAC2 to inhibit synaptic activity during other induced or pathological hyperactivity states, such as excitotoxicity, brain trauma, epilepsy, seizures, or neurological diseases, should be investigated in future studies.

## 4. Materials and Methods

### 4.1. Materials

AAC2 was synthetized and sterilized as described in [24,25], respectively. AAC2–hINS nanofibers were prepared as described previously [25]. Recombinant human insulin solution was purchased from Sigma-Aldrich (hINS, I9278, St. Louis, MO, USA).

### 4.2. Animal Study

The Institutional Animal Care and Use Committee of the Ohio State University (OSU) approved this animal study (protocol code 2007A0262-R5 and 26 January 2024). Wild-type (WT, C57BL/6J/Jax 000664) and APP/PS1 strain (APPswe, PSEN1dE9) 85Dbo/Mmjax/Jax 005864) mice were purchased from the Jackson Laboratory (Bar Harbor, ME, USA). Mice were fed a regular chow diet (Teklad LM-485 mouse/rat diet, irradiated; Envigo, Somerset, NJ, USA) under 12 h light/dark cycle. Mice were sacrificed by isoflurane inhalation followed by cardiac puncture.

### 4.3. Animal Study Design

The WT (*n* = 6) and APP/PS1 male mice (*n* = 33) from 4 to 8 weeks old underwent 1 week acclimation before the beginning of the study. Given that APP/PS1 male mice [90] develop glucose intolerance and insulin resistance earlier than APP/PS1 we selected APP/PS1 male mice for our pilot study as the most suitable model to test molecules that had previously alleviated these conditions in mouse models of diabetes in our earlier research [24,25]. Then, APP/PS1 mice were randomized into study groups (Figure 1a) based on weight and fasting glucose levels.

(1)WT, *n* = 6;(2)APP/PS1_control (C), *n* = 8;(3)APP/PS1_AAC2 (A), *n* = 9;(4)APP/PS1_hINS (I), *n* = 7;(5)APP/PS1_AAC2–hINS (A+I), *n* = 9.

For control, WT and APP/PS1 mice (C group) were injected into peritoneum (i.p.) every 2 d with sterile 10 µL PBS/g BW. A similar regimen was performed for all treatment groups of APP/PS1 mice injected with 10 µL PBS/g and AAC2 0.2 nmol/g BW for the A group; hINS 1.7 nmol/g BW for the I group; and 0.2 nmol of AAC2 bound with 1.7 nmol hINS/g BW for the A+I group of APP/PS1 mice. AAC2–hINS was prepared by mixing 200 µL of 0.1 mM AAC2 solution with 100 µL of hINS (10 mg/mL = 1.7 µmol/mL) into 700 µL of PBS and incubating for 30 min. The duration of study was 17 weeks. We followed the protocol developed for a mouse model of type 2 diabetes [25]; however, the decision to extend the study to 17 weeks was based on previous work by others [31,91] who described a slow, progressive amyloid plaque formation without significant cognitive decline in these mice. Body weight and fasting glucose were collected weekly. The enrollments into the other tests are shown in Figure 1a.

### 4.4. Metabolic Parameters: Fasting Glucose, Body and Brain Composition, and Activity Measurements

Mice were fasted for 4 h from 8 a.m.–12 a.m. to measure fasting blood collected by tail-tip pierce method using an Accucheck glucometer (Roche Diabetes Care, Mannheim, Germany). Body composition in living mice and a composition of frozen brains was measured with an EchoMRI™-100H Body Composition Analyzer for Live Small Animals (EchoMRI™, Houston, TX, USA). A Comprehensive Lab Animal Monitoring System (CLAMS, Columbus Instruments, Columbus, OH, USA) was used to measure energy expenditure, respiratory exchange ratio (RER), and XYZ movement activity by indirect calorimetry, as described before [24,25]. The measurements were performed on the individually housed mice at ambient temperature (22 °C), with 12 h light/dark cycles after treatment for 16 to 17 weeks.

### 4.5. RNA Isolation and NanoString mRNA Profiling

Mouse brains were dissected and sectioned longitudinally into two equal parts and stored at −80 °C. One half of frozen brain was powdered in liquid nitrogen. Homogenized brain powder (~10 mg) was used for mRNA isolation using the RNeasy Mini Kit (QIAGEN, Germantown, MD, USA). Subsequent quantification of the mRNA was carried out using ND-1000 UV/Vis Spectrophotometer (NanoDrop Technologies, Wilmington, DE, USA). mRNA (approximately 50 ng) with a 260/230 nm absorbance ratio > 1.8 and 260/280 nm absorbance ratio > 1.8 was used for profiling on the customized neurometabolic panel NanoString nCounter platform (CDR-ODZ_0622-19244 NanoString Technologies, South Lake Union, Seattle, WA, USA). Probes were designed to recognize mRNA of 99 neurometabolic genes quantitatively and specifically (Appendix A). These probes include 9 potential housekeeping genes and 20 AAC2-responsive genes related to neural function, which were identified in previous study in mouse models of T1DM [25].

### 4.6. NanoString Data Analysis

The processing of raw data, quality control measures, identification of housekeeping genes, pre-normalization visualization, normalization, post-normalization visualization, differential expression analysis, and functional analysis were conducted using R software (version 4.2.1), in accordance with the workflow delineated by Bhattacharya et al. [92]. Quality control (QC) for each sample adhered to specific parameters: imaging QC exceeded 75%, binding density QC ranged between 0.1 and 2.25, positive control linearity QC demonstrated an R^2^ value above 0.95, and the positive control limit of detection QC surpassed 2 standard deviations above the negative control’s mean. The selection of housekeeping genes and the quantity utilized for normalization was determined via the algorithm proposed by Vandesompele et al. [93]. Pre-normalization and post-normalization visualizations, critical for the detection of unwanted variances across all samples, were represented via relative log expression (RLE) plots [40] and principal component analysis (PCA) plots. The RUVSeq package (version 1.30.0) was employed for the normalization of NanoString raw data [94]. Differential expression analysis was executed using the DESeq2 package (version 1.36.0) [95]. The Kyoto Encyclopedia of Genes and Genomes (KEGG) and Reactome pathway over-representation analyses were carried out on the differentially expressed genes via the clusterProfiler (version 4.7.1.003) according to [96].

### 4.7. Behavioral Tests

All tests were performed at the Behavioral Core facility at OSU in blinded fashion using standard protocols. Mice were encoded and submitted for experiments 11–15 weeks after the beginning of study.

The open field test is a widely used method for evaluating locomotor activity and anxiety-related behaviors in unfamiliar environments [31]. The Open Field Photobeam Activity System (San Diego Instruments, San Diego, CA, USA) comprising a polypropylene light- and sound-protected open-field arena (36 cm × 36 cm) with two rows of infrared sensors mounted on the sides was used to detect horizontal and vertical movements of each mouse, distance traveled, time resting, number of rears, and time spent in the center versus the periphery. Activity counts were defined as interruptions in the infrared light sources by the animal.

The Rotarod test was employed to evaluate mice’s motor capabilities, including strength, balance, and coordination skills [97]. The duration each mouse stayed on the rotating barrel was automatically documented. The recording ceased when the animal either fell off or climbed to the top of the rod and inverted itself.

The elevated plus-maze test was used to evaluate anxiety-related behavior in response to a potentially threatening environment [31]. The elevated plus-maze test was performed using the ANY-maze tracking system (Stoelting, Wood Dale, IL, USA) with two open (exposed) arms (67 cm × 5.5 cm) and two closed and darkened (confined) arms (67 cm × 5.5 cm × 15 cm), where mouse behavior was recorded for 5 min. The number of entries and time spent in open and closed arms was collected.

The Barnes maze serves as a terrestrial evaluation of spatial learning and reference memory [98]. The Barnes maze test was carried out using the ANY-maze tracking system. Each mouse was placed in the middle of the maze at the start of each trial and allowed to explore for 2 min to memorize spatial cues around an elevated platform to locate a hidden goal box. Each trial ended when the mouse entered the escape box or after 2 min. Mice were allowed to spend 30 s in the box. Mice failing to find an escape box within 2 min were guided by the experimenter to the escape box to spend 30 s there. Boxes were cleaned to remove all cues between sessions. Each mouse received 3 trials per day over 5 consecutive days. Latency, distance traveled, and number of errors were recorded during training time. On day 8, the test trial was conducted by removing the escape box. Percent of time spend in this area was described as Path Q3. Mouse was placed in the middle of the maze and allowed to explore for a fixed interval of 90 s.

### 4.8. Statistics

All data were presented as mean ± SD. Survival curves were plotted with Kaplan–Meier method. Log-rank tests were performed for curve comparisons between groups. Statistical tests were performed with R software (version 4.2.1).

## 5. Conclusions

Growing evidence has shifted the paradigm of understanding AD pathology from late-phase amyloidogenic and NFT-mediated neurometabolic processes to early processes that result in fluctuating neurological function [99] and asymptomatic epileptiform activity associated with glucose hypometabolism [100]. These processes, occurring in the early phases of the disease, appear to define the cognitive performance and synaptic loss in AD patients [99] and, therefore, have become prime pharmacological targets.

The major outcome of this study is the evidence of two distinct synaptic gene expression patterns stimulated by INS and AAC2, which may drive divergent excitatory and inhibitory responses, respectively. The combination of AAC2 and INS into nanofibers blunted the effects of the individual constituents. Many questions remain unresolved in this proof-of-concept study, including the mechanism of ictogenesis; the pharmacokinetic profile of AAC2; the permeability of AAC2, INS, and AAC2–INS through the blood–brain barrier; and the long-term effects of chronic therapy with these compounds. Nevertheless, we demonstrated the neurodynamic effects of AAC2 in abolishing critical side effects of INS or modulating INS action overall, which may be desirable in some clinical scenarios. These findings could have implications for balancing E/I activity in early preclinical AD and could be explored in the context of other neurodegenerative conditions, including brain trauma, diabetes, and mental disorders characterized by excitatory and inhibitory imbalance, which pose risks for epileptiform activity, seizures, and cognitive decline.

## 6. Patents

O. Ziouzenkova, J.R. Parquette. US Patent App. 16/088,267 ‘Thermogenic compositions and methods’.

## Figures and Tables

**Figure 1 ijms-25-11689-f001:**
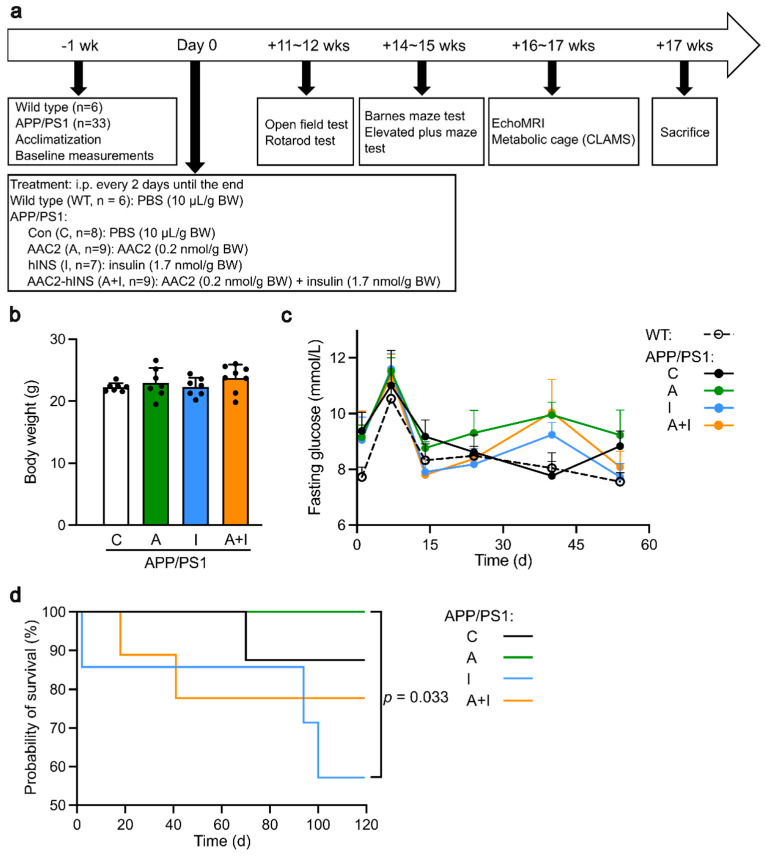
Experimental design and differential survival among WT mice and APP/PS1 control group (C) and groups treated with AAC2 (A), hINS (I), or AAC2–hINS (A+I) for 17 weeks. (**a**) Schematic diagram of treatment protocol of WT and APP/PS1 mice subjected to intraperitoneal injection of AAC2, hINS, or AAC2–hINS combination. Treatments were administered every 2 days until the end of the experiment. Premature death of mice was recorded. (**b**) Similar body weight of APP/PS1 mice from all groups prior to the treatment. Not significant (N.s.), ANOVA (**c**) Kinetics of fasting glucose concentrations in blood (mmol/L) in WT mice (dashed line, *n* = 7) and APP/PS1 mice treated with PBS (C, *n* = 7), AAC2 (A, *n* = 8), hINS (I, *n* = 6), and AAC2–hINS (A+I, *n* = 8) was measured for 54 days following treatment. N.s. ANOVA (**d**) Kaplan–Meier survival curve of APP/PS1 mice treated with PBS, AAC2, hINS, and AAC2–hINS throughout the whole experiment. *p* = 0.033 for comparison between AAC2 and hINS treatment via log-rank survival analysis.

**Figure 2 ijms-25-11689-f002:**
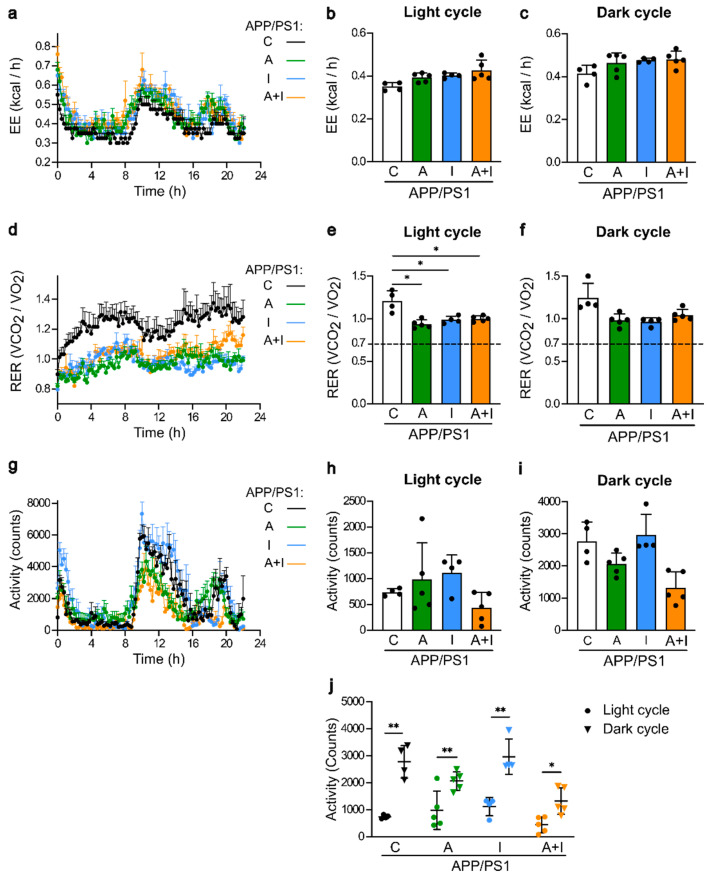
Similar effects of treatment with AAC2 (A, *n* = 5), hINS (I, *n* = 4), and their combination (A+I, *n* = 5) compared to control APP/PS1 mice treated with PBS (C, *n* = 4) on the metabolic status parameters and activity measured in metabolic cages during 24 h. (**a**–**c**) Kinetics of the energy expenditure (kcal/h) during 22 h (**a**), light cycle (**b**), and dark cycle (**c**). N.s., generalized linear model (GLM) [34] (**d**–**f**) Kinetics of the respiratory exchange rate (RER) during the same experiment during 22 h (**d**), light cycle (**e**), and dark cycle (**f**). Dashed line, the critical RER threshold at which carbohydrates begin to contribute to energy provision [35]. Asterisk, * *p* < 0.05, GLM. (**g**–**j**) The activity (XYZ) of APP/PS1 mice in same experiment during 22 h (**g**), light cycle (**h**), dark cycle (**i**), and a comparison of activity increase from a light to a dark period (**j**). Double asterisks, ** *p* < 0.001, paired *t* test.

**Figure 3 ijms-25-11689-f003:**
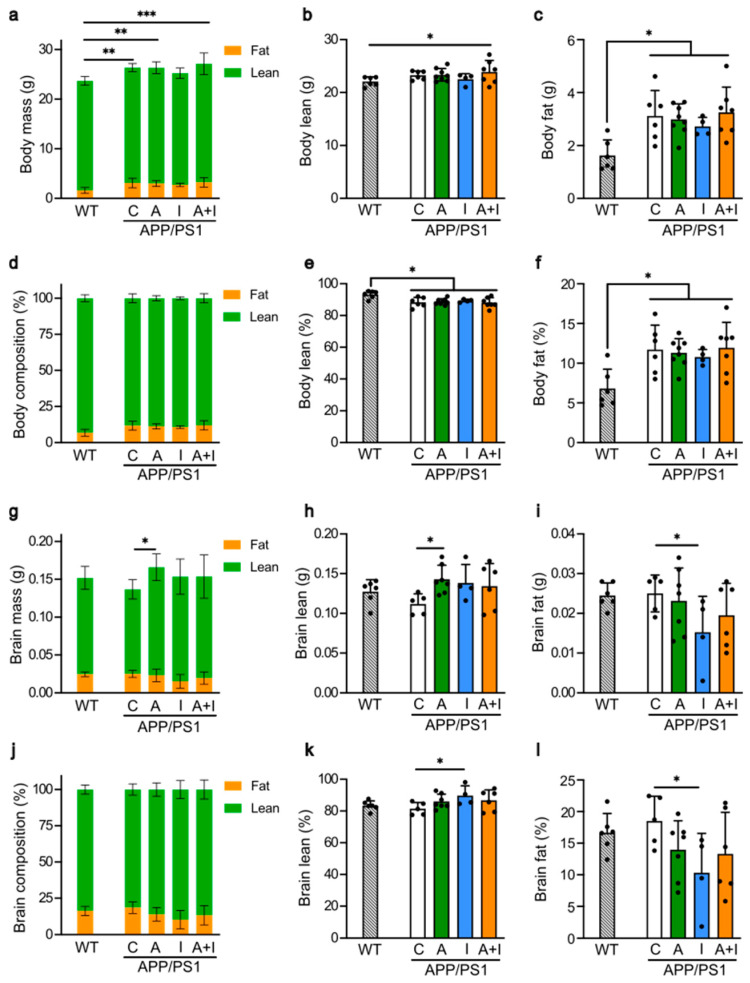
Treatment with AAC2 and INS altered brain mass and composition, respectively, without affecting body composition. Differences in body composition were measured in living WT (*n* = 6) and APP/PS1 mice (C, *n* = 6; A, *n* = 8; I, *n* = 4; A+I, *n* = 7) by EchoMRI: (**a**–**c**) The total body mass (**a**), body lean mass (**b**), and body fat mass (**c**) in of WT and APP/PS1 mice. Asterisks, comparison between WT and APP/PS1 mice: * *p* < 0.05, ** *p* < 0.01, *** *p* < 0.001. ANOVA (**d**–**f**) Body composition (**d**), lean (**e**), and fat (**f**) composition percentage were calculated taking into account mass of each animal. Asterisk, comparison between control and treated groups, * *p* < 0.05. ANOVA (**g**–**i**) Differences in brain composition were measured in frozen brains from same mouse groups by EchoMRI. The total brain mass (**g**), brain lean mass (**h**), and brain fat mass (**i**) of WT and APP/PS1 mice. Asterisk, * *p* < 0.05. ANOVA (**j**–**l**) The brain composition (% compared to mass of each animal) (**j**), brain lean (**k**), and brain fat compositions (%) (**l**) of WT and APP/PS1 mice. Asterisk, * *p* < 0.05. ANOVA.

**Figure 4 ijms-25-11689-f004:**
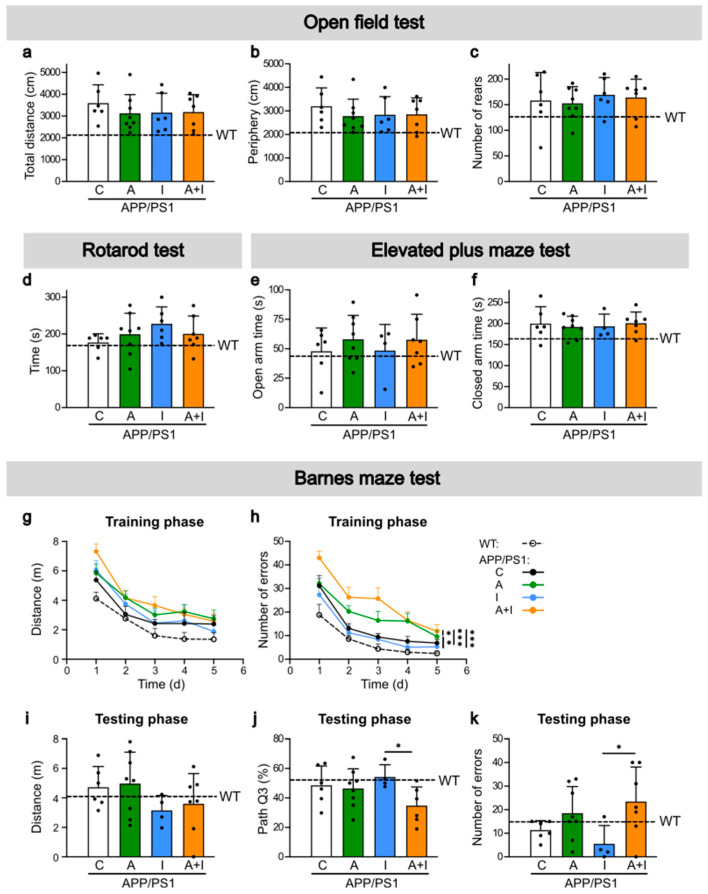
All APP/PS1 and WT mice exhibited similar cognitive characteristics, but learning outcomes differed between free and AAC2-bound INS-treated APP/PS1 mice. (**a**–**c**) Open field test was performed with WT (*n* = 6) and APP/PS1 mice after 11 weeks of treatment (C, *n* = 6; A, *n* = 8; I, *n* = 6; A+I, *n* = 7). Total distance (**a**), amount of activity spent in the periphery of the arena (**b**), and number of rears (**c**) were measured. N.s., ANOVA (**d**) Rotarod test was conducted in same animals after 12 weeks of treatment. N.s., ANOVA (**e**,**f**) elevated plus-maze (EPM) test was conducted on WT (*n* = 6) and APP/PS1 mice after 15 weeks of treatment (C, *n* = 6; A, *n* = 8; I, *n* = 4; A+I, *n* = 7). The amount of time the mice spent in the open (**e**) and closed arms (**f**) was measured, N.s., ANOVA. (**g**–**k**) Barnes maze test was conducted on WT (*n* = 6) and APP/PS1 mice after 14 weeks of treatment (C, *n* = 6; A, *n* = 8; I, *n* = 4; A+I, *n* = 7). During the training phase (Day 1–5), the total distance travelled by mice (**g**) and the number of errors, which were visits made to holes other than the one that leads to the goal box (**h**), were measured. The probe trial was for 90 s. Asterisks represent comparison between INS and other treatment groups, AAC2 vs. A+I, and C vs. A+I ** *p* < 0.01, and *** *p* < 0.001, repeated ANOVA. During the testing phase, the total distance (**i**), the proportion of distance travelled within Q3 area containing the escaping hole (**j**), and the number of errors (**k**) were measured. Asterisks represent comparison between free INS and bound with AAC2/INS nanofibers, * *p* < 0.05, ANOVA.

**Figure 5 ijms-25-11689-f005:**
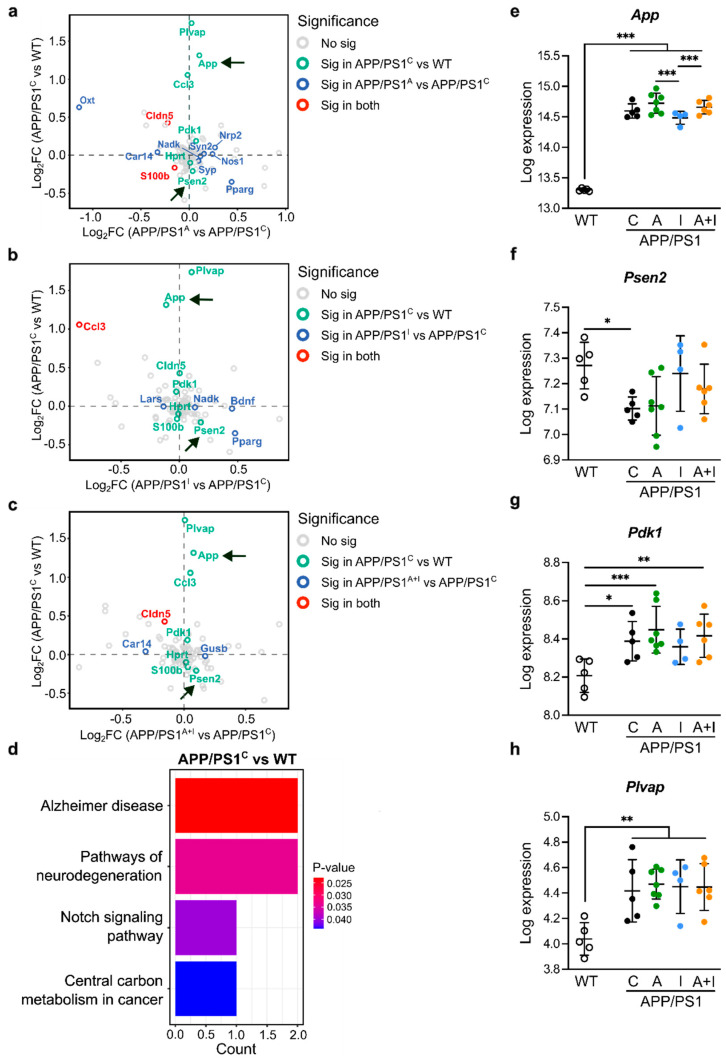
Treatment with AAC2, hINS, and their combination did not influence the expression of canonical AD genes in the brains. (**a**–**c**) Scatter plot compared differently expressed genes (DEGs) between APP/PS1 control and WT (arrows, canonic genes, such as *App* and *Psen2*) on Y axes with DEGs between treatment and control groups in APP/PS1 mice on X axes: (**a**) AAC2 treatment vs. control; (**b**) hINS treatment vs. control, (**c**) AAC2–hINS treatment vs. control. (**d**) KEGG pathway over-representation analysis of the DEGs between APP/PS1 control and WT mice. (**e**–**h**) The expression of cerebral *App* (**e**), *Psen2* (**f**), *Pdk1* (**g**), and *Plvap* (**h**) in WT (*n* = 5) and APP/PS1 mice C, *n* = 5; A, *n* = 7; I, *n* = 4; A+I, *n* = 6). Differential analysis was performed using DESeq2 R package mainly based on the GLM and empirical Bayes shrinkage, * *p* < 0.05, ** *p* < 0.01, *** *p* < 0.001.

**Figure 6 ijms-25-11689-f006:**
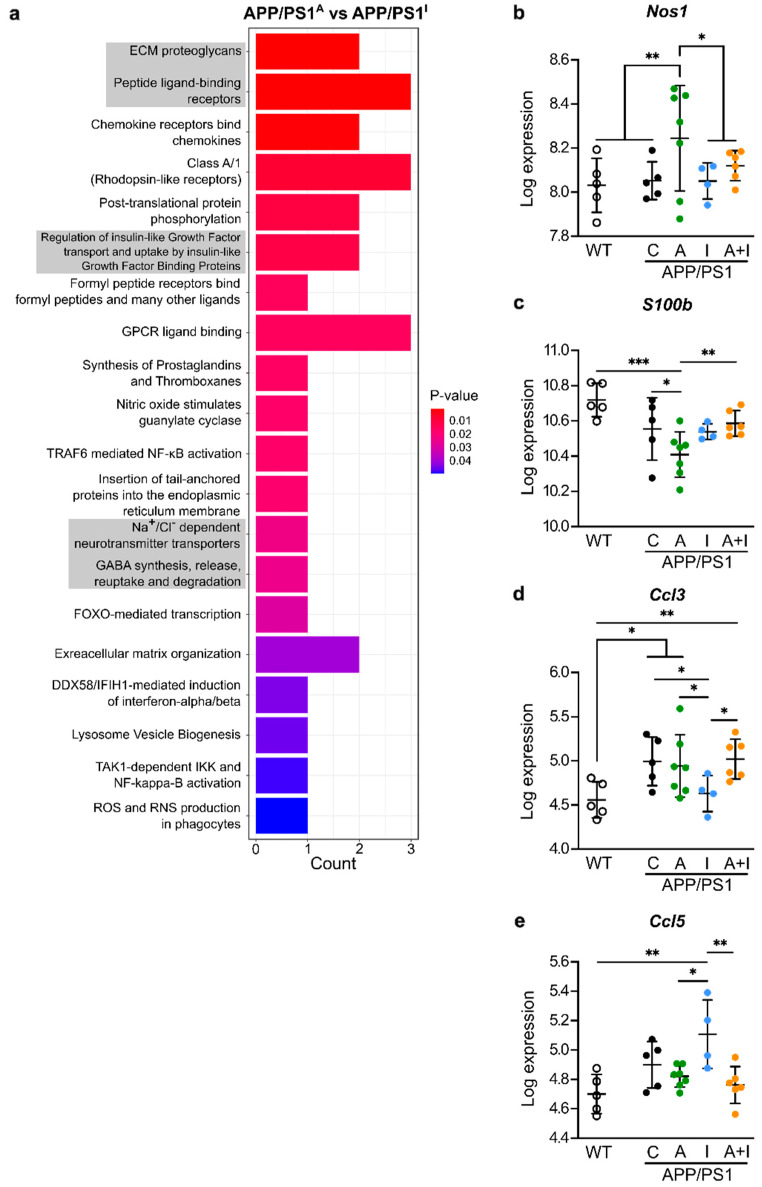
AAC2 and hINS regulate different neuroinflammatory pathways in APP/PS1 mice. (**a**) Reactome pathway over-representation analysis of the DEGs between WT and AAC2- and hINS-treated APP/PS1 mice. (**b**–**e**) The expression of cerebral *Nos1* (**b**), *S100b* (**c**), *Ccl3* (**d**), and *Ccl5* (**e**) in WT (*n* = 5) and APP/PS1 mice (C, *n* = 5; A, *n* = 7; I, *n* = 4; A+I, *n* = 6). Differential analysis was performed using DESeq2 R package mainly based on GLM and empirical Bayes shrinkage * *p* < 0.05, 0.001 ≤ ** *p* < 0.01, *** *p* < 0.001.

**Figure 7 ijms-25-11689-f007:**
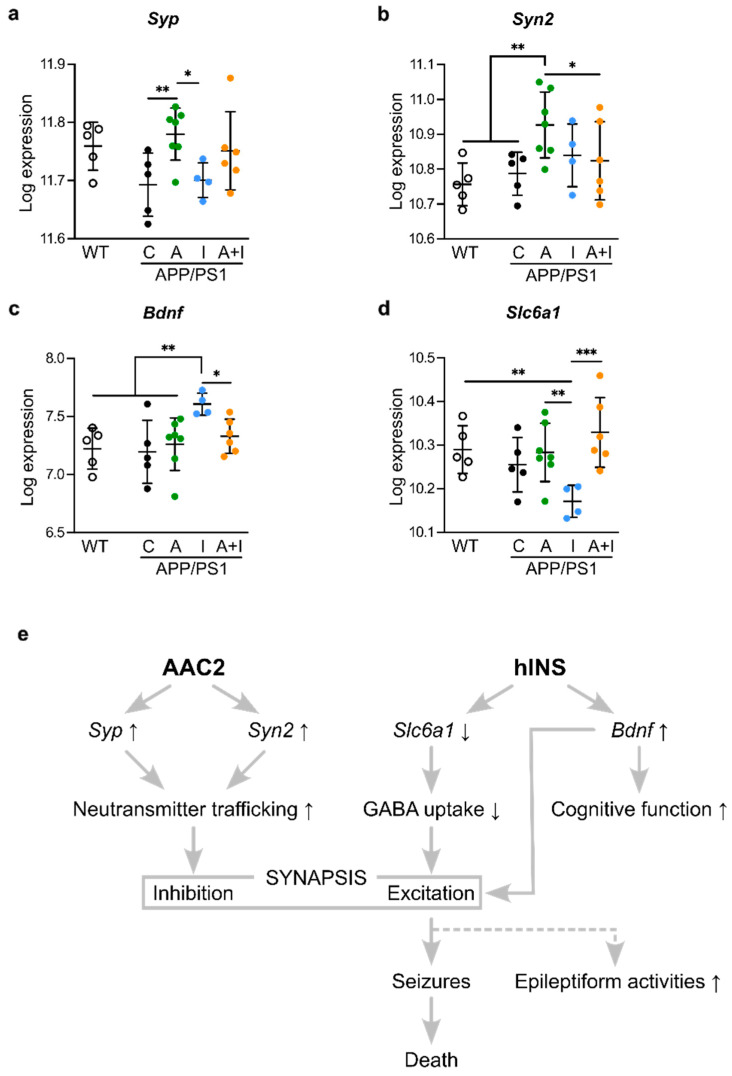
Different responses of APP/PS1 mice to AAC2 and hINS result from the possible disbalance of GABAergic synaptic excitation and inhibition (**a**–**d**) The expression of cerebral *Syp* (**a**) *Syn2* (**b**), as well as implicated in excitation *Bdnf* (**c**), *Slc6a1* (**d**) in WT (*n* = 5) and APP/PS1 mice (C, *n* = 5; A, *n* = 7; I, *n* = 4; A+I, *n* = 6). Differential analysis was performed using DESeq2 R package mainly based on the GLM and empirical Bayes shrinkage, * *p* < 0.05, *** p* < 0.01, *** *p* < 0.001. (**e**) Hypothesized mechanism for discrepant effects of AAC2 and INS treatment in APP/PS1 mice.

## Data Availability

The original contributions presented in this study are included in the article/Appendix A. Further inquiries can be directed to the corresponding author(s).

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
