# Peer review of "Amino Acid Compound 2 (AAC2) Treatment Counteracts Insulin-Induced Synaptic Gene Expression and Seizure-Related Mortality in a Mouse Model of Alzheimer’s Disease"

_ijms, 2024, doi:10.3390/ijms252111689_

Round 1
Reviewer 1 Report
Comments and Suggestions for Authors
The article by Deng et al. investigated the effects of AAC2, a previously reported lysine dipeptide with a coumarin side chain, on synaptic gene expression, seizure activity, and mortality in an Alzheimer's disease (AD) mouse model. The study compares AAC2 treatment to insulin and a combination of AAC2 and insulin, using APP/PS1 mice. Their study revealed found that insulin treatment increases seizure-related mortality and alters synaptic gene expression whereas AAC2 treatment promoted the expression of Syn2 and Syp synaptic genes, leading to preserved brain composition and higher survival. Overall, I think this manuscript is very well structured, the experimental design and analysis are robust. I only have 1 major suggestion.
In this paper, the authors reported that the seizure-related mortality was prevented with combined use of insulin and AAC2, however, they did not fully explain how AAC2 counteracts insulin-induced seizures at the molecular level. Could the authors provide a possible molecular explanation or at least a hypothesis in the discussion section?
Author Response
Reviewer 1.
The article by Deng et al. investigated the effects of AAC2, a previously reported lysine dipeptide with a coumarin side chain, on synaptic gene expression, seizure activity, and mortality in an Alzheimer's disease (AD) mouse model. The study compares AAC2 treatment to insulin and a combination of AAC2 and insulin, using APP/PS1 mice. Their study revealed found that insulin treatment increases seizure-related mortality and alters synaptic gene expression whereas AAC2 treatment promoted the expression of Syn2 and Syp synaptic genes, leading to preserved brain composition and higher survival. Overall, I think this manuscript is very well structured, the experimental design and analysis are robust. I only have 1 major suggestion.
In this paper, the authors reported that the seizure-related mortality was prevented with combined use of insulin and AAC2, however, they did not fully explain how AAC2 counteracts insulin-induced seizures at the molecular level. Could the authors provide a possible molecular explanation or at least a hypothesis in the discussion section?
We would like to thank this Reviewer for the encouraging suggestion to speculate on the mechanism. Several studies suggest the inhibitory impact of leptin via its interaction with the leptin receptor, leading to phosphorylation of ERK1/2 and increased expression of Syp and Syn2. The atypical binding of AAC2 to the leptin receptor in our previous studies led to robust activation of the ERK pathway, which is a plausible mechanism for the suppression of Syp and Syn2 and the absence of seizures in APP/PS1 mice treated with AAC2 or AAC2/INS. We have added this hypothetical mechanism at the end of the Discussion section (Lines 485-492):
‘It is well established that the inhibition of GABAergic neurons depends on the leptin receptor (LepR) [88]. Moreover, chronic leptin administration activates ERK1/2 in neonatal rats [89]. Therefore, activation of the LepR/ERK1/2 pathway by AAC2 [24] could trigger an inhibitory response in these neurons through increased expression of Syp and Syn2. Although mechanistic studies were beyond the scope of this paper, the role of AAC2/LepR signaling in the induction of Syp and Syn2 appears to be a plausible mechanism supporting inhibitory synaptic balance and preventing early death in APP/PS1 mice. Regardless of interpretation, our results revealed that AD pathology could increase the risk of side effects of INS, which could be reduced by the formation of AAC2-INS nanofibers’.
Reviewer 2 Report
Comments and Suggestions for Authors
Current report is going to highlight the potential of amino acid-compound 2 (AAC2) in regulating synaptic gene expression in Alzheimer’s disease (AD) and insulin-induced contexts related to seizure activity. Please conduct the concerns below.
1. Background of AAC2 needs to introduce in detail. Then, rationale of current study is easier to follow.
2. The used male APP/PS1 mice also must describe in clear. Merit of the mice in current study was not mentioned.
3. Study design including the duration of 17 weeks, injection every 2 days, and the effective dose of each product must follow the established report.
4. Animal Monitoring System needs reference(s) to support.
5. Limitation of Nanostring neurometabolic analysis seems ignored. Why?
6. In line 359, opposite regulation of synaptic gene by INS and AAC2 must follow the previous report(s).
7. Merits of the combination of AAC2 and INS into nanofibers were not mentioned in a good way. Please improve it.
Author Response
Reviewer 2.
Current report is going to highlight the potential of amino acid-compound 2 (AAC2) in regulating synaptic gene expression in Alzheimer’s disease (AD) and insulin-induced contexts related to seizure activity. Please conduct the concerns below.
We would like to thank the Reviewer for the thorough review and helpful comments.
Background of AAC2 needs to introduce in detail. Then, rationale of current study is easier to follow.
We added more information (Lines: 80-84, 86-90) about the effects of AAC2 and its nanofibers with insulin, as well as provided the rationale for the current study, in the paragraph before the last in the Introduction section.
- The used male APP/PS1 mice also must describe in clear. Merit of the mice in current study was not mentioned.
Thank you. We have added the justification for using male mice in Methods section 4.3 (Lines: 513-517).
Given that APP/PS1 male mice develop glucose intolerance and insulin resistance earlier than APP/PS1 females {Li, 2016 }, we selected APP/PS1 male mice for our pilot study as the most suitable model to test molecules that had previously alleviated these conditions in mouse models of diabetes in our earlier research {Lee, 2021; Lee, 2020 }.
- Study design including the duration of 17 weeks, injection every 2 days, and the effective dose of each product must follow the established report.
We added the description of this protocol development in Methods section 4.3 (Line: 545).
- Animal Monitoring System needs reference(s) to support.
We have provided the new references in Methods section 4.4 (Lines: 530-533).
- Limitation of Nanostring neurometabolic analysis seems ignored. Why?
The recent lawsuit against NanoString, which resulted in a $31 million damages award, centered on allegations that NanoString's GeoMx Digital Spatial Profiler infringed on patents held by 10x Genomics. However, this is a different technology from what we use in our paper. We utilize the original technology described by Geiss et al. (2008), which was also applied in our previous studies. Additionally, we validated our findings with gene expression data from APP/PS1 brains and provided an assessment of morphological and behavioral changes, correlating them with the identified pathways from our gene expression analysis. As stated in the revised paper (Lines 416-418):
The quantitative assessment of gene expression by Nanostring is a reliable tool for identifying candidate pathways {Geiss, 2008 ;Shen, 2018}; however, it provides limited insights into the signal transduction underlying epileptiform or synaptic activity in response to INS, AAC2, and their combination’.
- In line 359, opposite regulation of synaptic gene by INS and AAC2 must follow the previous report(s).
We address this request in Section 2.7. the last sentence (Lines: 358-362): ’The gene expression analysis demonstrates the opposite regulation of synaptic gene by INS and AAC2 consistent with previous report on their opposite effects on GLUT1 glucose uptake in human brain barrier endothelial cells {Lee, 2020 #93}, and in agreement with earlier findings on the dependence of synaptic inhibition in cortical pyramidal neurons and thalamic relay neurons on this pathway {Rajasekaran, 2022 #153}’
.
- Merits of the combination of AAC2 and INS into nanofibers were not mentioned in a good way. Please improve it.
To emphasize the importance of AAC2-INS nanofibers, we have added an additional statement at the end of the Discussion (Lines: 491-492; 630-632):
Regardless of interpretation, our results indicate that AD pathology may increase the risk of side effects from INS, which could be mitigated by the formation of AAC2-INS nanofibers.
We stated in the Conclusion section (at the end of the Methods) that:
Nevertheless, we demonstrated the neurodynamic effects of AAC2 in abolishing critical side effects of INS, or modulating INS action overall, which may be desirable in some clinical scenarios.